# Hepatitis D Review: Challenges for the Resource-Poor Setting

**DOI:** 10.3390/v13101912

**Published:** 2021-09-23

**Authors:** Alice U. Lee, Caroline Lee

**Affiliations:** 1Concord Repatriation General Hospital, University of Sydney, Sydney, NSW 2139, Australia; 2Hepatitis B Free, Sydney, NSW 2139, Australia; 3Westmead Hospital, Sydney, NSW 2139, Australia; caroline.carrie.lee@gmail.com

**Keywords:** hepatitis D, hepatitis B, resource limited

## Abstract

Hepatitis D is the smallest virus known to infect humans, the most aggressive, causing the most severe disease. It is considered a satellite or defective virus requiring the hepatitis B surface antigen (HBsAg) for its replication with approximately 10–70 million persons infected. Elimination of hepatitis D is, therefore, closely tied to hepatitis B elimination. There is a paucity of quality data in many resource-poor areas. Despite its aggressive natural history, treatment options for hepatitis D to date have been limited and, in many places, inaccessible. For decades, Pegylated interferon alpha (Peg IFN α) offered limited response rates (20%) where available. Developments in understanding viral replication pathways has meant that, for the first time in over three decades, specific therapy has been licensed for use in Europe. Bulevirtide (Hepcludex^®^) is an entry inhibitor approved for use in patients with confirmed viraemia and compensated disease. It can be combined with Peg IFN α and/or nucleos(t)ide analogue for hepatitis B. Early reports suggest response rates of over 50% with good tolerability profile. Additional agents showing promise include the prenylation inhibitor lonafarnib, inhibitors of viral release (nucleic acid polymers) and better tolerated Peg IFN lambda (λ). These agents remain out of reach for most resource limited areas where access to new therapies are delayed by decades. strategies to facilitate access to care for the most vulnerable should be actively sought by all stakeholders.

## 1. Introduction

Hepatitis delta virus (HDV), discovered over 40 years ago by Mario Rizzetto [1], is a small, enveloped RNA virus. HDV is a defective or satellite virus that requires hepatitis B virus (HBV), particularly the expression of the hepatitis B surface antigen (HBsAg), for its life cycle. The HDV virion consists of a shared envelope protein with HBV, and a ribonucleoprotein containing the viral genome of circular single-stranded RNA [2]. HDV enters hepatocytes through the same mechanism as HBV due to their shared envelope protein [3]. This is an important therapeutic target for cell entry inhibitors.

HDV infection causes the most severe disease of the hepatitis viruses, with higher likelihood of progression to cirrhosis and hepatocellular carcinoma (HCC) compared with HBV monoinfection. However, much of its epidemiology, natural history and treatment options are still limited. There are approximately 250 million people worldwide with hepatitis B [4], and approximately 5% of HBsAg-positive individuals are estimated to have HDV [4]. Worldwide prevalence of HDV has been estimated in three meta-analyses, with results ranging from 12 million people [4] to 60–72 million [5,6]. HDV is estimated to be attributed to 18% of cirrhosis and 20% of HCC associated with HBV [4].

Globally, rates of HDV have decreased with improved hepatitis B vaccination. However, geographic hotspots with pockets of high prevalence exist in Central Asia, Oceania, West and Central Africa, the Amazon Basin of South America, and Eastern Europe. Prevalence varies widely within and between countries. There is inaccessibility to HDV diagnostics, especially HDV RNA, in many regions such as Asia and Africa, limiting estimation of HDV prevalence [7].

Universal testing for HDV in HBsAg-positive individuals is not widely recommended or available, contributing to likely underestimates of its true prevalence. Guidelines from Europe (EASL) and the Asia Pacific Region (APASL) recommend testing for HDV for all HBsAg-positive individuals [8,9]. However, American guidelines (AASLD) recommend screening in key populations at risk of HDV such as those with risk factors or born in high endemic regions, or HBsAg-positive individuals with low or undetectable HBV DNA but elevated ALT levels [10]. Antibody to HDV is the mainstay of screening for HDV infection, with subsequent confirmation of active HDV infection by HDV RNA testing for those with positive antibodies. However, access to quality testing is still limited and costly.

Pegylated interferon α has been the only available treatment approved for use in hepatitis D, with response rates of approximately 20%. Approval of the entry inhibitor bulevirtide in Europe promises increased response rates. However, more information is required to determine how it can best be used. Treatments for HDV remains inaccessible for resource-limited settings due to a lack of studies and prohibitive costs as well as intensive monitoring requirements. There is increasing attention to the need to address gaps in care for people with HBV and HDV, and development for the treatment and care of those coinfected in the resource-limited setting.

In this review, we highlight the epidemiology, natural history, diagnostics, and treatment options for HDV, with a focus on resource-limited settings.

## 2. Epidemiology

Approximately 250 to 300 million people worldwide are living with hepatitis B infection, of whom 5% have been estimated to have HDV-antibody positivity, equating to 12 million people worldwide with HDV [4]. Reported prevalence of HDV is higher amongst people attending liver clinics [4]. However, higher global prevalence estimates of 13–15% have been reported in earlier meta-analyses, correlating to 48–60 million or 72 million people with HDV, respectively [5,6]. Given the widespread lack of testing and data, HDV is likely underestimated and exact figures are unknown.

Increased coverage of HBV vaccination since the 1990s has contributed to reduced rates of HDV amongst younger populations in regions of the world such as Europe, North Africa, Saudi Arabia and Israel [7,11]. However, even with improved HBV vaccination, pockets of HDV are likely to persist within countries due to variable intra-country vaccination coverage, particularly in rural and remote areas.

Geographic variation in HDV prevalence is well recognised [4,5,6]. However, HDV prevalence does not exactly parallel trends in HBV prevalence. Despite significant burden of hepatitis B in China, India and Indonesia, low rates of hepatitis D (including none at all in Indonesia) have been reported [5]. Low rates are reported from high-income regions such as the USA, where prevalence of HDV is 0.36% [12]. Even in areas of low prevalence, higher rates are seen in at-risk groups. A study of over 25,000 veterans with positive HBsAg found 3.4% prevalence of HDV, and associations with HBV/HCV coinfection and substance abuse [13]. American guidelines do not recommend routine testing for hepatitis D [10]. Key populations at higher risk of HDV include people who inject drugs (PWID) [4,5], people living with HIV and HCV, people with high-risk sexual behaviours including MSM [6,14], and haemodialysis recipients [4]. Higher rates of HDV also persist amongst immigrants from endemic regions.

Higher rates of HDV are seen in low- and middle-income countries in Africa, Central and South Asia, Oceania and the Amazon basin of South America. Meta-analysis by Stockdale and colleagues found that the regions with highest prevalence of HDV amongst HBsAg-positive individuals were Mongolia, the Republic of Moldova, and Western and Middle Africa [4]. This is based on limited data which are often old. Pockets of high prevalence in the Pacific islands has also reported with more than half the HBsAg-positive people in Kiribati testing positive HDV antibody, of whom 73% were viraemic (HDV RNA positive) [15].

Despite the lack of comprehensive prevalence data, the burden of HDV is high amongst many low- and middle-income regions of Africa, Asia and Oceania where hepatitis B prevalence often exceeds 5% and hepatitis B vaccination coverage is suboptimal. In Africa, HDV prevalence varies between regions and has been described as “clusters of endemnicity” [14,16]. Meta-analyses of HDV prevalence in North Africa reported pooled prevalence of 5% amongst the general population and 21% amongst people with liver disease [16]. Meta-analysis of HDV prevalence in Sub-Saharan Africa found pooled prevalence of 7%, 26%, and 0.05% in West, Central, and East and Southern Africa, respectively [14]. Most studies did not use confirmatory methods for HDV antibody positivity, either by retesting or HDV RNA testing; it is likely HDV RNA would not have been available in most settings [14].

In Central Asia, Mongolia reports some of the highest rates of HDV of 40–60% amongst HBsAg-positive patients [17]. Other central Asian countries with high rates of HDV reported amongst cirrhotic patients include Uzbekistan, Tajikistan, and Krygyzstan. Meta-analysis of HDV prevalence in the WHO Eastern Mediterranean region of 62 studies showed a weighted mean HDV prevalence of 15%, up to 37% amongst people with chronic hepatitis, cirrhosis and HCC [18]. In Pakistan, rates of 50–60% HDV positivity amongst HBsAg-positive individuals attending liver clinics was reported in several districts, with variation noted and higher prevalence in central rural regions [19]. In Iran, variation in prevalence of HDV between different regions was described in a meta-analysis of 19 studies. In one study in the south Iran, prevalence of HDV amongst patients with cirrhosis was as high as 20%. The authors suggested difference in transmission risk factors amongst lower socioeconomic regions in the South may have contributed to this variation, but highlighted that blood transfusions were a major source of infection [20]. Similar patterns of intra-country variability in prevalence have been reported in Vietnam, the Mikayo islands and Okinawa of Japan and Yakutia in Russia. Geographic hots spots of higher prevalence of HDV are also reported in Romania and Moldova [7].

Prevalence rates of hepatitis D are highly variable in the Pacific islands, from as high as 55% in Kiribati, 22% in Nauru and none reported in other islands including Vanuatu, Fiji and Tonga, where small samples of hepatitis B-positive patients have been screened for hepatitis D (unpublished data) [15,21].

In most of Central and South America, rates of chronic hepatitis B (CHB) are low (approximately 1%), except for the Western Amazon Basin in Brazil, Venezuela, Columbia, Peru and Ecuador, where both HBV and HDV are endemic. A meta-analysis of HDV prevalence in South America estimated a pooled prevalence of 22% amongst HBsAg-positive individuals [22]. However, authors noted significant selection bias, with most studies from the Amazon basin and inclusion of old studies from the 1980s to 1990s, and suggested these figures were unlikely nationally representative [22].

Prevalence data for HDV are limited, contributing to under-recognition of the need for hepatitis D testing. With improved awareness, and development of testing and care guidelines, a roadmap can be developed to better understand the burden of disease and develop strategies to manage this serious disease.

## 3. Natural History

Acute hepatitis D infection can occur as a co-infection with HBV and HDV, or as superinfection with HDV infection on existing chronic hepatitis B [23]. Acute coinfection presents as acute hepatitis with elevated transaminases with higher risk to fulminant hepatitis compared to acute hepatitis B monoinfection. The pattern of hepatitis is often biphasic with the first flare due to hepatitis B, and the second due to hepatitis D [23]. It results in spontaneous recovery in the majority of cases (95% in immunocompetent individuals) with the remainder progressing to chronic coinfection. The rates of progression to chronic HDV infection are similar to acute hepatitis B monoinfection [2,3]. Positive anti-HBc IgM supports the diagnosis of acute coinfection.

Superinfection occurs when a person with chronic hepatitis B is infected with HDV. In asymptomatic hepatitis B patients, it can lead to acute hepatitis and may be the first time a diagnosis of hepatitis B is made, whilst in patients with chronic active hepatitis B, the acute flare can lead to hepatic decompensation [2]. Unlike acute coinfection which often self-resolves, 90% of superimposed hepatitis D infection leads to chronic HDV infection. Hepatitis D is the most common viral cause of fulminant hepatitis either as acute coinfection or superinfection and has a poor prognosis with high rates of mortality if transplantation is not available [24].

HDV usually causes suppression of HBV replication in approximately 70% of cases, with HBV-HDV co-dominance in 28% and a less commonly HBV dominant (3%) [6]. Fluctuations in replication dominance can occur, [25] as well as spontaneous fluctuations in HDV RNA, including spontaneous HDV RNA undetectability [26]. This suppression can lead to anti-hepatitis B surface Ab positivity and hepatitis B e antigen negativity, with HDV replication being the main factor affecting disease severity. In some cases, spontaneous HBsAg loss can occur. Persistence of hepatitis B viraemia is associated with more severe disease [27].

Less commonly, latent infection after liver transplantation has also been reported. This is characterised by the presence of anti-HDV in the liver, absence of HBsAg and HDV RNA in the blood and is mostly associated with mild disease [23].

The natural history of chronic hepatitis D ranges from asymptomatic carriage or mild symptoms to severe disease. People with HDV in general are a decade younger in age than those with HBV monoinfection. Chronic hepatitis D results in a more severe clinical course than chronic hepatitis B monoinfection and is the most severe form of chronic viral hepatitis [2], with more than 3-fold higher risk of cirrhosis (OR 3.84) [6], 2-fold higher risk of decompensation and liver transplantation, and 2-fold higher risk of mortality [28]. On average, progression to cirrhosis and HCC occurs within 5 years and 10 years, respectively [6].

Higher rates of cirrhosis represent an indirect increased risk of HCC without evidence as to the direct oncogenicity of the virus. Hepatitis D association with HCC was initially controversial, but more recent studies indicate that HDV is associated with a higher risk of HCC than HBV monoinfection, though exact mechanisms are not fully understood [4]. Meta-analysis performed by Alafaiate and colleagues (2020) demonstrated a significant association of HDV with HCC (pooled odds ratio 1.28) despite significant study heterogeneity [29]. HDV is associated with a 3-fold higher HCC risk and 2-fold higher mortality compared with HBV infection alone [3]. A higher risk of HCC was associated with HIV infection (pooled odds ratio 7.13) but not with HCV infection in meta-analyses [29].

Although detailed pathogenesis of HDV-induced liver disease is not known, viral factors and host immune response factors affect progression of disease. HDV virus is thought to be directly cytopathic during acute infection whilst host immune mediated response dominates injury in chronic infection [30]. Factors that affect progression of HDV infection include superimposed versus acute coinfection, genotype, high levels of HDV viral replication, and coinfection with HIV and HCV [24]. In cases of triple infection with HCV, hepatitis D is typically the dominant infection [31].

## 4. Diagnosis

The first-line screening test for HDV is HDV antibody (IgM and IgG) in hepatitis B-positive patients, followed by HDV RNA testing to confirm ongoing viraema. Positive HDV RNA for more than 6 months indicates chronic infection.

Acute coinfection is characterised by positive HBsAg, anti-HBc IgM, anti-HDV IgM and HDV RNA. The ALT increase can occur in a biphasic pattern (2–5 weeks apart) with the first flare due to hepatitis B and the second due to hepatitis D. The antibody response to HDV is slower than that of HBV. Acute HDV infection may be missed as anti-HDV positivity can occur late. Acute infection usually resolves within 2–10 weeks [32]. Additionally, following resolution of infection, antibodies to HDV can disappear over time so that past infection may not be detected. This can contribute to difficulty estimating the true prevalence of HDV.

In patients presenting with acute hepatitis in endemic areas, efforts should be made to differentiate acute coinfection from superinfection in established CHB, as this influences prognosis and management. Acute coinfection is characterised by the presence of anti-HBc IgM and anti-HDV (IgM). Presence of anti-HBc IgG and anti-HDV (IgM) with negative anti-HBc indicates superinfection. The HDV antibody response can be slow with delay up to weeks. HDV IgG is seen in the convalescence phase. HDV RNA should then be performed to confirm viraemia. A negative test should be repeated whilst other causes of hepatitis should also be screened. Liver biopsy is occasionally the only positive diagnostic test in a small cohort of patients with severe hepatocellular inflammation and necrosis reported with a definitive diagnosis available with the identification of HD antigen on biopsies [33]. Routine use of liver biopsy is not recommended in resource-limited settings.

For people with known CHB with an acute monophasic flare of ALT, presence of HDV IgM and positive HDV RNA indicates HDV acute superinfection. This can lead to suppression of HBV DNA and in some cases, clearance of hepatitis B, or both viruses (10%). HDV typically suppresses HBV so that HDV RNA levels are high whereas HBV DNA levels are low or undetected. HBeAg is negative in more than 80% of cases. Shifts in the dominant virus can occur over time. Hepatitis D antigen cannot be identified on ELISA as it is in complex with anti-HDV and hence has limited diagnostic utility in practice [23].

Chronic HDV infection occurs when there is positive total anti-HDV longer than 6 months with a positive HDV RNA. Persistence of positive HDV RNA carries a poorer prognosis, associated with progression to cirrhosis and liver decompensation [34], and is a predictor of mortality [35]. Challenges in diagnosis are associated with development of anti-HDV during later stages as well as the lower sensitivities of the assays. Despite improved viral load testing sensitivities (with lower limit of detection to 14 IU/mL) and HDV RNA standardisation developed by the WHO and the Paul Ehrlich institute, laboratory sensitivities and dynamic range variations continue. None of these diagnostic tools are readily available in the resource-poor setting.

## 5. Treatment

Antiviral therapy is not indicated for acute hepatitis D infection. Supportive care and referral for transplantation if available in the case of acute liver failure are recommended.

In patients with a positive anti-HDV, irrespective of viral load, assessment of liver disease staging should be undertaken using locally available noninvasive, or where required, invasive (liver biopsy) tools. Noninvasive liver staging tools used in the setting of HDV include the aspartate aminotransferase (AST) to platelet ratio index (APRI) and the fibrosis-4 score (FIB4), as well as measures of liver stiffness with transient elastography [36,37]. If HDV RNA is negative, ongoing monitoring is required with view to repeat testing as clinically indicated. Consideration for hepatitis B treatment should be made based on liver disease staging based on viral load and ALT according to local guidelines. However, treating hepatitis B does not affect hepatitis D as ongoing HDV replication proceeds independent of HBV replication.

In comparison to the complex treatment algorithms for CHB monoinfection, all patients with positive viral load for hepatitis D should be considered candidates for therapy, irrespective of disease staging. Disease activity (with ALT) should be assessed to monitor disease as well as treatment response. Special considerations should be taken for those with decompensated liver disease where current therapies are not indicated.

The ultimate end goal of treatment is clearance of HBsAg with seroconversion to anti-HBsAg (functional cure) but remains elusive in most cases [38]. Hence, other defined response to treatment includes biochemical (normalisation of ALT) and virological response (undetectable HDV RNA). Sustained virological response (SVR) is defined as undetected viral load 6 months post completion of therapy. However, high rates of relapse are reported beyond this time frame, and ongoing monitoring is recommended. Routine use of histological response (both histological defined as improvement in inflammation (>2 points) and fibrosis (1 point) as well as elimination of HDV RNA) is not recommended.

Off-label use of weekly subcutaneous pegylated interferon α for 48 weeks has been the mainstay of HDV treatment for the past 20 years with a SVR of 20–30%. Its mechanisms of action include antiviral and immunomodulatory effects. In addition to the low response rates and late relapses in approximately half, requirement for close monitoring and side effects limit its use [39]. Combing therapy with nucleos(t)ide analogues did not improve overall outcomes (HIDIT-1 and -2) [40,41]. Extending therapy to 96 weeks was not associated with improvement in virological response. For those that did achieve response, improved clinical outcomes with lower rates of cirrhosis and lower overall mortality compared with non-responders is seen [40].

In 2020, the first antiviral agent for hepatitis D was approved under conditional authorization by the European Medical Agency for patients with compensated liver disease and positive viraemia with or without nucleos(t)ide analogueshere [36,42]. Bulevirtide (Hepcludex^®^) 2 mg subcutaneous daily stops HDV and HBV from entering the human hepatocyte allowing for recovery, protecting uninfected cells and allowing potential clearance. Bulevirtide (BLV) is an entry inhibitor, a lipopeptide that mimics the sodium taurocholate transporting polypeptide (NTCP) receptor binding domain of the L-HBsAg, thereby competing with the HBsAg for attachment to the entry receptor NTCP, mechanism shared by HDV virus. This receptor is also used for bile acid transport.

Initially developed as Myrcludex-B, in a phase II MYR203n study, three subcut doses (2, 5 and 10 mg per day) was assessed as mono therapy or in combination with 48 weeks of pegylated IFN α. The best outcome was seen in the cohort who received combination treatment and 2 mg, with normalisation of ALT and undetected HDV viral load in 53.8% and 53.3%, respectively at 72 weeks (24 weeks after stopping therapy). Further, 40% achieved a 1 log reduction in HBsAg. Side effects from pegylated IFN α were mild, with BLV side effects related to increased bile acids so that monitoring bile acid levels is recommended on treatment [43]. Long-term follow up studies are required. Earlier studies of combination with tenofovir with different doses of BLV for 24 weeks did not demonstrate any benefit by the addition of tenofovir with undetected HDV viral load at end of treatment in 46, 47 and 77% of patients treated with 2, 5 or 10 mg per day, respectively and 3% on tenofovir monotherapy [43]. Similar improvements in ALT were seen in 43, 50 and 40% of those on BLV and 6% on tenofovir. High rates of relapse after stopping therapy were seen in all groups.

Small studies have also shown its effectiveness at a higher dose in patients, one with decompensation receiving 10 mg of bulevirtide for up to 3 years. Biochemical and virological response maintained for 3 years without relapse or breakthrough despite reduction in dose. This is associated with improvement in clinical response from decompensation to compensated liver disease [44].

Current recommendations on duration of treatment are unclear. Treatment discontinuation can be considered after 6 months of normal ALT if abnormal at baseline, and undetectable HDV associated with HBsAg seroconversion. Kinetics modelling suggests 3 years of monotherapy may lead to 50% long-term off-treatment response despite ongoing positive HBsAg [45].

Studies are currently underway in other countries and assessing its potential role in the treatment of chronic hepatitis B infection. Ongoing real-life data are keenly awaited.

Combination treatment with pegylated IFN α offers a higher SVR, but is not well tolerated and, in many, contraindicated. Hence, viral suppression as the target of monotherapy with BLV is considered for those with no other alternatives and advanced disease. Requirement for daily injections, need for regular blood test monitoring to check platelet counts and bile acid levels, and current recommendations suggesting that treatment is managed in expert centres further limits accessibility to many vulnerable populations and is contradictory to ongoing efforts to increase access to care and decentralising hepatitis services. With increased real-life data and safety profiling, roll out to resource-poor setting using a simplified algorithm should be explored. Patients in such areas have no other opportunity for care and would otherwise come to adverse outcome, and when potential harm of treatment is weighed against its benefit, the balance would in many cases be well in favour of treatment.

## 6. Newer Treatments under Evaluation

Drugs that inhibit other steps in viral life cycle show promise. Following entry into the hepatocytes, HDV (L-HDV Ag) interacts with hepatitis B protein to form infectious particles. The prenylation step by cellular farnesyltransferase is essential for viral assembly, replication and subsequent infection of other hepatocytes. Lonafarnib is an oral prenylation inhibitor shown to reduce HDV viral load. Combined with ritonavir which enables a 4–5-fold increase in systemic exposure and improved gastrointestinal tolerability, these well tolerated oral agents offer on treatment viral suppression with minor gastrointestinal side effects. ALT normalisation and viral suppression are seen at 6 months on therapy with 50 and 100 mg of lonafarnib with or without ritonavir. No significant changes or decline in HBsAg is seen. [46] When combined with pegylated interferon alpha for 12 or 24 weeks, ALT normalisation and reduction in viral suppression are improved at 78 and 89%, respectively [46,47,48].

The nucleic acid polymer REP-2139 and its bioequivalent variant REP-2165 block HBV subviral particle release from the hepatocyte, a mechanism that is shared by hepatitis D in its release. A small study of 12 cirrhotic patients assessed 15 weeks of 500 mg weekly intravenous infusion, followed by another 15 weeks of REP-2139 and pegylated IFN alpha, followed by pegylated IFN alpha alone for another 33 weeks. High rates of side effects and ALT flare were reported. Viral suppression was seen in 10 of 12 patients as well as half cleared HBsAg, which was sustained up to 3 years after treatment [49,50]. Its potential role in HDV therapy is under further evaluation.

Lambda interferon is highly expressed in hepatocytes and hence use of pegylated IFN λ may lead to improved safety profile and lower rates of side effects when compared with pegylated interferon alpha. At doses of 180 ug per week in addition to nucleos(t)ide analogues, ALT normalisation and viral suppression were seen in up to 36% at 72 weeks with fewer side effects. When combined with lonafarnib and ritonavir for 24 weeks, 42% had undetected viral load at end of therapy, and 19% with undetected viral load 24 weeks off therapy [48]. Dose reduction and treatment discontinuation were required in 11 and 15%, respectively.

Combination therapy requires ongoing exploration with agents that target different stages of the virus life cycles combined with immune modulators.

## 7. Summary and Recommendations

Hepatitis D is a highly aggressive viral hepatitis which can rapidly progress to cirrhosis and liver cancer. It remains under-recognised, underestimated, and under-treated. Given recent suggestions that the burden of disease is higher than previously recognised, and with the advent of new therapeutics, now is the time to focus attention to this dangerous virus. Through laboratory support, health worker training, public health policies and funding, resource-poor settings should be empowered to provide care for people affected by hepatitis D. Preventative hepatitis B vaccination requires ongoing attention, as well as prevention of mother to child transmission. Improved understanding of local transmission patterns will help support preventative interventions such safe injection practices, including culturally sensitive targeting local practices involving blood exposure.

In hepatitis D-endemic areas, all patients with HBsAg should be screened with anti-HDV. For those with positive HDV antibodies, linkage to care is recommended irrespective of the HDV viral load. Liver disease staging should be assessed using locally available noninvasive tools (such as APRI, FIB4 scores, or vibration-controlled transient elastrography). A liver biopsy is not mandatory.

In areas where prevalence data are limited, all HBsAg-positive patients with advanced liver disease should be screened with anti-HDV. High-risk groups should also be screened for hepatitis D. Testing should be performed in clinical scenarios suggestive of HDV infection including acute hepatitis flare in known HBsAg-positive patients, or when acute coinfection is clinically suspected.

The paucity of data in many regions highlights that recommendations on testing needs revision as local epidemiology is better understood. This is key to guide allocation of already scarce resources and should be considered early as hepatitis B treatment programs are being rolled out in some of the most resource-limited settings [51,52].

Until recently, complex and costly hepatitis programs meant that care of patients in resource-poor settings was limited to diagnosis and management of its complications. However, with more affordable diagnostics and antiviral agents, and simplified treatment algorithms, both hepatitis B and C treatment programs are being introduced to some of the most resource-poor settings. These programs use various models, with particular focus on decentralization, and can be utilised for hepatitis D care. Screening programs for hepatitis B and C are underway throughout the Pacific Islands. Niue has successfully completed its “cure a country” whole-of-population screening program, with the majority of the population undergoing hepatitis B and C testing [53]. Hepatitis B programs, including access to antivirals, have been introduced in Kiribati where high rates of HDV is well documented [52]. Similar programs are underway in Papua New Guinea (PNG), Fiji, Vanuatu, Solomon Islands and Tonga. National health ministries are investing to support laboratory systems including viral load testing in-country, supported by regional laboratory (Victorian Infectious Diseases Reference Laboratory [VIDRL]), procurement pathways, training and expansion of hepatitis care. Sampling of a small number of hepatitis B-positive cases for hepatitis D (HDV Ab followed by reflex viral load testing) is currently underway by VIDRL but remains limited. Longer-term testing strategies need development, as do guidelines for screening and care of hepatitis D.

Access to treatment for hepatitis D, despite its numerous challenges, should not be discarded as impossible. Historically, access to care in resource-limited settings is often delayed with many countries still unable to access nucleos(t)ide analogues despite its well established value for over two decades. Surely, given this experience, proactive measures should be considered to prevent this inequitable delay. Strategies such as telehealth to support local health workers, increased transportation of samples, and appropriate models of care should be considered in these settings. The newest agent BLV appears promising, with a good tolerability profile even in those with advanced disease. Given that the alternative is inevitable aggressive disease, access should be prioritised. Engaging pharmaceutical companies to appeal to social responsibility with tiered costing is needed. This should ideally be performed at high level by policy makers, supported by civil society advocacy groups globally.

For now, all patients with HBV–HDV coinfection and detectable HBV viral load should be considered for hepatitis B antiviral therapy, irrespective of HDV viral load.

Treatment with pegylated interferon α or BLV is reasonable in some regions where access to trained personnel and laboratory services is available. Other therapies for cancers and immune disease are already underway and expertise available locally. In patients without advanced liver disease, 48 weeks of combination therapy as per European guidelines would be reasonable. In those with advanced disease and where support from medical team is limited, BLV monotherapy should be considered as the treatment of choice given its better side effect profile. Cost is the major barrier.

Patients on combination treatment should be monitored every two weeks for the first two months and thereafter monthly. Viral load testing should be performed at 48 weeks and then at 74 weeks.

For patients on BLV monotherapy, monthly bloods followed by second monthly bloods is suggested. Therapy should be long-term with annual viral load, HBsAg and evaluation for ongoing need for treatment. Where possible, treatment should be considered for three years duration. Patients who relapse post treatment cessation should be recommenced on treatment.

Where treatment is not available or indicated, three-monthly clinical review and ALT measurement are suggested. Care of patients with advanced liver disease is recommended as per local guidelines.

Despite the challenges in resource-poor settings, research in these areas should be prioritised and considered for inclusion in clinical studies, particularly phase 3 and 4 studies. If implemented, this may support resource development for both staff and infrastructure.

Training health workers in screening is a priority, and should be harmonized throughout the region considering available resources and local prevalence, to ensure clear guidelines are communicated and revised regularly as new information becomes available.

Elimination goals for viral hepatitis are ambitious and of increasing attention. The Global Health Sector strategy for Viral Hepatitis 2015–2021 did not specifically address hepatitis D [54]. We cannot achieve global goals of eliminating viral hepatitis by 2030 without addressing gaps in hepatitis D care.

## 8. Key Summary of Recommendations

Prevalence studies for HDV should be considered integral to establishment of hepatitis programs.In areas with limited prevalence data, all patients with advanced liver disease and at-risk populations should be assessed for HDV Ab.In endemic areas, all patients with HBsAg should be screened with anti-HDV. Linkage to care is recommended for all anti-HDV positive patients.Patients with an acute flare of hepatitis should be considered for screening for anti-HDV.All patients with coinfection, irrespective of viral load and treatment status, should be linked to care and monitored.Where HDV treatment is available but funding is limited, priority should be given to those with advanced liver disease.All patients with HBV-HDV coinfection with detected HBV DNA should be considered for HBV antiviral therapy irrespective of staging.All patients with positive HDV RNA should be considered for HDV treatment.All cirrhotic patients with CHB should be treated with nucleos(t)ide analogues.Discussions should be undertaken to explore avenues to provide access to BLV.Combination therapy with pegylated interferon and BLV for 48 weeks may be considered for those without advanced liver disease. This should only be undertaken where trained medical supervision and access to regular laboratory assessment are available.Monotherapy with BLV should be considered for those with advanced liver disease.

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
