# Peer review of "Hepatitis D Review: Challenges for the Resource-Poor Setting"

_viruses, 2021, doi:10.3390/v13101912_

Round 1

Reviewer 1 Report

This is an interesting and well-written review. However, some issues needs to be addressed prior to its acceptance, especially in terms of combined therapy of BLV and pegIFN.

Abstract

- “and where indicated” There is a typo and the comas are missing: and, when indicated,

- “ Response rates of over 50% achieving sustained viral suppression as well as 24 good tolerability profile has been reported.It may be added that those are preliminary results.

Introduction

- Page 1, line 38: “surface antigen HBsAg” introduce the acronym HBsAg.

- Page 1, line 42: “higher likelihood of progression” do you mean higher than HBV monoinfection or with the highest rate of progression among liver viruses?

- Page 2, line 52: “Central Asia, Oceania, 52 and West and Central Africa” I suggest to add Amazon Basin and Eastern Europe to this list.

- Page 2, line 59: “for all those with chronic HBV” I suggest to modify the sentence to “all those HBsAg positive individuals ”or “all those with chronic HBV, regardless of the phase of the infection” in order to emphasize the universal recommendation for all HBV patients.

- Page 2, line 60: “key populations at risk of HDV” I suggest to add “such as those with risk factors or born in regions with high endemicity”

- Page 2, line 63: “HDV RNA viral loading testing” this expression is redundant, use HDV RNA testing or viral load testing.

Epidemiology

- Page 2, line 88: “Geographic variation in HDV prevalence is well-recognised” I suggest to assess the addition of a figure regarding estimated HDV prevalence.

- Page 3, line 104: “This is based on limited data which is often old data” omit the last data or change by “which dates from the 90s”.

- Page 3, second paragraph: it may be stressed the important limitation of data from the studies performed in Africa, concerning both HDV antibodies and especially HDV RNA.

Natural history

- I suggest to add the usefulness of anti-HBc Ig M to discriminate between coinfection and superinfection in patients with acute hepatitis.

- Page 4, line 167 “Hepatitis D is the most common viral cause of fulminant hepatitis either as acute coinfection or superinfection and has a poor prognosis with high rates of mortality if transplantation is not available.(26)”- I suggest to move this sentence after this one “Unlike acute coinfection which of ten self-resolves, 90% of superimposed hepatitis D infection leads to chronic HDV infection.” Since the text afterwards (“HDV usually causes suppression...”) refers to chronic HDV infection, I suggest beginning at a new paragraph and move this information to the paragraph concerning chronic HDV infection (page 4, line 174).

- Page 4, line 166 “In some cases, HBsAg loss can occur.” I suppose authors refer to spontaneous HBsAg loss.

- Page 4, line 163 “Fluctuations in replication dominance can occur” I suggest to add that spontaneous fluctuations on HDV RNA have also been described, including spontaneous HDV RNA undetectability [Palom A et al. Aliment Pharmacol Ther 2021].

- Page 4, line 175 “People with HDV in general are a decade younger in age” than HBV monoinfected individuals?

Diagnosis

- Throughout the diagnosis and treatment text: change “anti HDV Ab” by anti-HDV

- Page 5, line 203 “and HDV RNA PCR” the term PCR can be omitted.

- Page 5, line 204 “Presence of anti-HBc IgG and anti HDV Ab (IgM)” add “and negativity for anti-HBc Ig M.

- In this section, it may be added that the achievement of persistently undetectable HDV RNA, either spontaneously or after treatment, positively impact on prognosis of the disease [Palom A, Aliment Pharmacol Ther 2020; Romeo F et al Gastroenterology 2009; Niro GA, et al.  J Hepatol. 2010].

- Since the paper in focus in limited resource settings, it would be interesting the addition of some data on alternative diagnostics tests such as dried-blood spot, e.g. Martinez-Camprecios J, et al. J Virol Methods 2021.

Treatment

- Page 5, line 237 “if available” in case of acute liver failure.

- Page 5, line 242 “liver disease staging as well as viral load and ALT” change by “liver disease staging based on viral load, ALT and assessment of liver fibrosis and necroinflammation”

- Page 5, line 247 “Disease activity (with ALT) should be assessed to monitor disease as well as treatment response.” As well as quantitative HBsAg and HDV RNA if available.

- Page 5, line 250 “with seroconversion to anti HBsAg” this part can be omitted since to achieve function cure only HBsAg lost is required, regardless of anti-HBs seroconversion.

- Page 5, line 253 “complete or partial > 1 log IU/ml decline”substitute by “Undetectable HDV RNA or ≥2 log10 HDV-RNA decline”.

- Page 5, line 253 “Sustained virological response (SVR) is defined” I suggest to avoid the term SVR in patients with chronic hepatitis Delta, and just use virological response, due to the high rate of late relapse.

- Page 6, first paragraph: avoid the use of the term SVR; Add a sentence on the usefulness of qHBsAg and HDV RNA decline for guiding pegIFN treatment.

- Concerning the EMA authorization, two issues should be addressed: The EMA authorization for Bulevirtide is conditional; The EMA recommendation for Bulevirtide is as monotherapy or in co-administration with a nucleos(t)ide analog, not with pegIFN.

- Regarding the study from ref. 37, it should be highlighted that data on long-term follow-up after stopping therapy are lacking.

Newer treatment under evaluation.

- Page 7, line 315 “Drugs that inhibit other steps in viral life cycle show promise. Following entry into the hepatocytes, HDV (L-HDV Ag) interacts with hepatitis B protein to form infectious particles.” It would be interesting the addition of a picture showing the mechanism of action in the HDV life cycle of the new drugs.

- Page 7, line 330: “weekly infusion” substitute by weekly intravenous infusion

- Page 7, line 361: “staging should be assessed using locally available noninvasive tools” there is no prior mentioned at the text on noninvasive tools, and it would be usefull its addition at the diagnostic section, focusing on the proposed cut-off for HDV [Takyar V, et al. Aliment Pharmacol Ther 2017;45(1):127-138; Lutterkort GL, et al. Liver Int 2017 ;37(2):196-204]

Summary and Recommendations

- This section is a bit long; I would suggest its revision and reduction, especially the paragraph from line 390 to 401.

Page 8, line 364 “At risk groups” do you mean high risk groups?

- Page 8, line 365 “Testing should be done in clinical scenarios suggestive of HDV infection including acute hepatitis flare in known HBsAg positive patients, or in cases where acute coinfection is clinically suspected.”  The most prevalent scenario for diagnosis of chronic HDV is screening of patients with chronic HBV infection, especially in case of increase ALT levels.

- Page 8, line 402 “For now, all patients with hepatitis B detectable viral load should be considered for hepatitis B antiviral therapy, irrespective of HDV viral load.” HBV treatment criteria in patients with HDV are the same than for monoinfected patients, so this sentence should be rewritten.

 - Page 8, line 404 “The introduction of pegylated interferon α in combination with BLV is reasonable in some regions where access to trained personnel and laboratory services are available.” The recommendation of the EMA is BLV alone or combined with NUCs. Evidence is lacking to recommend the combination of pegIFN and BLV so far. In this line, recommendation number 9 and 11 may be revised.

- Something is missing in recommendation 12.

Author Response

Dear Reviewer,

Thank you for your consideration and review of our manuscript. In the attached document, we detail responses to the comments in the attached file.

Kind regards,

Caroline Lee

on behalf of Alice Lee, Caroline Lee

Reviewer 2 Report

In this review paper, the authors summarize epidemiology, natural history, diagnosis and treatment of hepatitis D virus (HDV) infection, focusing on the diagnosis and care for HDV infection in the resource limited settings especially. Because HDV infection could be underdiagnosed, the improvement of access to tests and therapies is important. This paper is organized well and covers important topics. I have some specific comments to be addressed as below.

  1. Page 8, lines 402-403. The description “all patients with hepatitis B detectable viral load should be considered for hepatitis B antiviral therapy” is not common. Is this meant only for patients with HDV coinfection/superinfection? If so, please state it clearly with an appropriate reference. It is the same for lines 445-446, page 9.
  2. Page 6, line 288. “BVT” should be corrected to “BLV”.
  3. Page 6, line 285 and page 9, line 451 and 460. “BVL” should be corrected to “BLV”.
  4. Page 8, line 382. Please show a full-spelling of “PNG”.

Author Response

Dear Reviewer,

Thank you for your consideration and review of our manuscript.

In the attached file, we respond to the specific comments.

Kind regards,

Caroline Lee

on behalf of Alice Lee, Caroline Lee

Round 2

Reviewer 1 Report

Authors have properly addressed all the concerns from this reviewer. Just few points need further revision:

Summary and Recommendations

- Page 8, line 386: there is a type “withmore”, substitute by with more.

- Page 9, line 417: “The introduction of pegylated interferon α in combination with BLV is reasonable in” since currently there is neither data nor recommendation for the use of this combination, I suggest changing this sentence, for instance “Treatment with pegylated interferon α or BLV is reasonable in”

- Page 9, recommendation 11: I understand the authors’ rational to keep the recommendation as stated in the original version of the manuscript. However, I recommended changing “should” by “may” in view of the scarce evidence and long-term efficacy of the combination of pegIFN plus Bulevirtide.

Author Response

Dear Reviewer and Editors,

Thank you for your review of our amendments and comments. We agree with the three recommendations and have made further amendments accordingly.

  • Page 8, line 386: there is a type “withmore”, substitute by with more. Agreed, amendment made
  • Page 9, line 417: “The introduction of pegylated interferon α in combination with BLV is reasonable in” since currently there is neither data nor recommendation for the use of this combination, I suggest changing this sentence, for instance “Treatment with pegylated interferon α or BLV is reasonable in”. Agreed, amendment made
  • Page 9, recommendation 11: I understand the authors’ rational to keep the recommendation as stated in the original version of the manuscript. However, I recommended changing “should” by “may” in view of the scarce evidence and long-term efficacy of the combination of pegIFN plus Bulevirtide. Agreed, amendment made.

Thank you for your ongoing review of our manuscript.

Kind regards,

Alice Lee & Caroline Lee